# Therapeutic hypothermia after out of hospital cardiac arrest improve 1-year survival rate for selective patients

Ofir Koren [1,2]*, Ehud Rozner[1], Sawsan Yosefia[3], Yoav Turgeman[1,2]

**1** Heart Institute, Emek Medical Center, Afula, Israel, **2** Bruce Rappaport Faculty of Medicine, Technion-Israel Institute of Technology, Haifa, Israel, **3** Internal Medicine C, Emek Medical Center, Afula, Israel

* Ofirko1@clalit.org.il

**Data Availability Statement:** All relevant data are in the paper and its Supporting Information files.

**Funding:** The authors received no specific funding for this work.

## Abstract

### Background

Therapeutic Hypothermia (TH) is a standard of care after out-of-hospital cardiac arrest (OHCA). Previous reports failed to prove a significant benefit for survival or neurological outcomes. We examined whether the proper selection of patients would enhance treatment efficacy.

### Method

We conducted a retrospective cohort study. Data was collected from January 2000 and August 2018. Patients were enrolled after OHCA and classified into two groups, patients treated with TH and patients who were not treated with TH.

### Results

A total of 92 patients were included in the study. 57 (63%) patients were in the TH Group and 34 (37%) in the Non-TH group. There was no statistical difference in favorable neurological outcomes between the groups. Patients presenting with ventricular fibrillation had a higher 1-year survival rate from TH, while patients with asystole were found to benefit only if they were younger than 65 years (p < .007, p < .02, respectively).

### Conclusion

Therapeutic Hypothermia patients failed to demonstrate a significant benefit in terms of improved neurological outcomes. Patients treated with TH following ventricular fibrillation experienced the most benefit in terms of 1-year survival, while patients who had suffered from asystole experienced a modest benefit only if they were younger than 65 years of age. Guidelines should address age and primary arrhythmia for proper treatment selection.

**Competing interests:** The authors have declared that no competing interests exist.

# 1. Introduction

Cardiac Arrest is a sudden cessation of cardiac activity as a result of ventricular fibrillation, asystole, or pulseless electrical activity [1]. The most common cause of cardiac arrest is ischemic heart disease [2–3]. The survival rate of cardiac arrest is very low, with a mortality rate of more than 90%. The only intervention therapy that has been studied so far in randomized trials is therapeutic hypothermia [4–5].

Therapeutic hypothermia was presented to the world in the early 1950s following initial reports of clinical benefits, both in terms of survival and neurological outcomes, in several patients who experienced cardiac arrest. The theoretical basis for starting the treatment was physiological and pathophysiological mechanisms that until now have not been fully understood [6–9].

During cardiac arrest, the blood supply to the body's organs decreases significantly and even stops. Global ischemia transforms the energy processes in the body into anaerobic processes, leading to the secretion of harmful oxidative products such as free radicals, amino acids, and inflammatory products. When blood flow is restored, both by Cardiopulmonary Resuscitation (CPR) and by Return of Spontaneous Circulation (ROSC), the byproduct of long-term ischemia adversely affects cardiac tissues, which are the main oxygen consumers. This leads to irreversible damage. Hypothermia, in this approach, is designed to reduce oxygen consumption of the core organs, reduce free radicals, and protect cell membranes to prevent intracellular oxidation [7–10].

In 2015, the American Cardiology Association, the European Cardiology Association, and the International Liaison Committee on Resuscitation (ILCOR) recommended the use of TH for comatose patients who gained spontaneous circulation after cardiac arrest due to shockable rhythm (class II, LOE A). In non-shockable rhythm cases, such as asystole or PEA, the benefit of therapeutic hypothermia was less evidenced (class II, LOE B) [11–14].

In the following years, several studies supported TH use, while others failed to prove a direct link between treatment and outcomes. Some of the studies regarded the cost-benefit of this complex treatment [15–22].

In 2016, a large meta-analysis of six prospective studies involving approximately 1399 patients found no significant benefit in either neurological outcomes or survival rate from TH use. The limitations of the study concerned mainly patient selection. Several trials included a high proportion of asystole or pulseless electrical activity patients. Patients with cardiac arrest following non-shockable rhythm were represented in small groups in previous studies, and the treatment efficiency was less pronounced in those groups. The broad diversity of target core temperatures among study trials also affected study validity [23].

## 1.1 Rational of the study

Our institute has been performing mild therapeutic hypothermia since 2008. The goal of the study was to examine its effectiveness in both neurological and survival aspects in subgroup patients.

# 2. Methods

We designed a retrospective cohort study conducted at our intensive cardiac care center. The study population included patients hospitalized after out-of-hospital cardiac arrest (OHCA) from January 2000 to August 2018. The patient's medical information was collected from the hospital's computer systems and Clalit health data service (Orion, Ofek, and Chameleon). Information regarding the events prior to hospital admission was collected from on-site family

members and emergency medical staff (EMS). Medical records from an external defibrillator were collected and analyzed.

Patients eligible for the study (S1 Table. Inclusion and exclusion criteria) were divided into two groups according to the therapeutic approach: patients who were treated with TH (TH-Group) and patients who were not treated (Non-TH Group). Since 2008, TH has been the standard of care in OHCA patients. Therefore, we used a randomized, demographic-matched group of patients from 2000 to 2008 as a control group and assessed treatment efficacy.

TH was initiated in the intensive cardiac care unit with the use of the thermoregulation system Criticool, manufactured by MTRE Advanced Technology Ltd, Israel. The cooling system was connected to a garment to facilitate cooling water over all body surfaces. We also use iced water gastric lavage cooling via a nasogastric tube and the continuous IV infusion of cooled isotonic saline solutions. The target cooling temperature was set to 33˚C, and the target re-warming temperature was set to 36˚C.

The primary endpoint was defined as the rate of survival of patients during hospitalization and up to one year after discharge. The secondary endpoint was divided, focusing on several points, the first being the rate of major adverse cardiovascular events (MACE), including acute myocardial infarction, congestive heart failure, arrhythmia, cardiac arrest, and sepsis within 30 days and up to one year after discharge. An additional secondary endpoint was the neurological status according to the CPC score during hospitalization, after 30 days, and up to one year following discharge [24].

## 2.1 Ethics

The study was approved by the Ethics Committee of the hospital in accordance with the Helsinki Convention No. EM-0062-17. Informed consent was not required due to the confidentiality of patent data.

## 2.2 Research planning

During the study period, 238 patients were hospitalized in our ICCU due to OHCA after regaining spontaneous circulation. 48 patients did not meet the inclusion criteria and two patients were excluded due to the discontinuation of treatment. 190 patients were eligible for study and were divided into two groups based on therapeutic approach. 57 were treated with hypothermia, and 133 were not. We randomly selected a group of 35 patients who were not treated with hypothermia, matched by a demographic propensity score, to serve as a control group (Fig 1. Research Planning).

## 2.3 Statistics

Considerations in calculating the sample size were based on several key points: incidence of OHCA, primary outcome, and efficacy of therapeutic hypothermia. We estimated an inpatient mortality rate of 30% after OHCA and treatment efficacy among the patient groups of about 10% with a at least 15% group difference. Sample size calculation estimated a total of 88 participants in a 2:1 ratio for 95% CI and 80% power. Differences in the two groups' demographic data were tested by a $\chi 2$ test or Fisher exact test where appropriate for the categorical data and by a student's t-test or the Mann-Whitney U test in the case of non-parametric data for continuous data. The treatment group was divided into subgroups by age ($<65$ / $\geq 65$) and primary arrhythmia (VF/asystole). Kaplan Meier survival analysis was performed in order to test differences in one-year survival time between the control group and the four treatment subgroups (age group by primary arrhythmias). Propensity score matching was performed using logistic

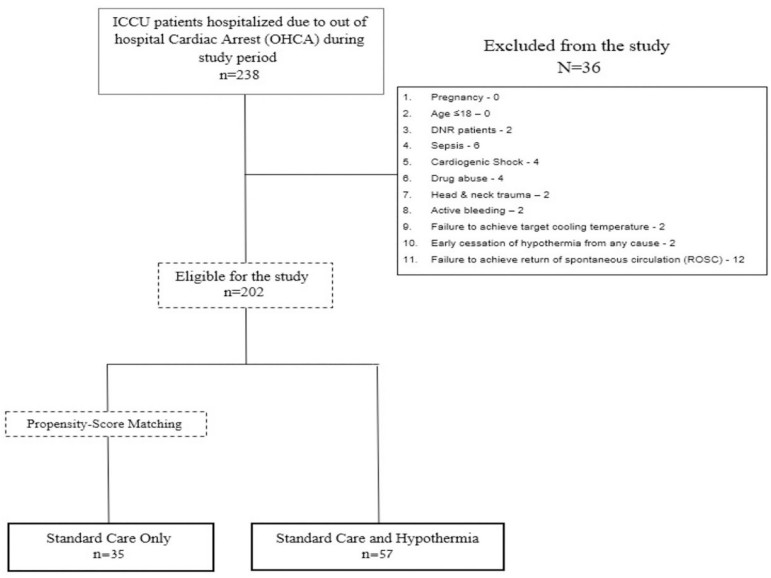

**Fig 1. Research planning.**

regression analysis and the greedy matching technique. Calculation was done using both R (MatchIt) and SAS for optimal bipartite matching. Cox regression analysis was performed in order to assess the predictors for 1-year mortality. Statistical analysis was done using SAS software version 9.4 and R (MatchIt). Statistical significance was obtained if p < .05.

## 3. Results

The study included 92 patients in total: 57 patients in the TH-group and 35 patients in the Non-TH group. The groups, as designed, did not differ in demographic characteristics such as age, gender, cardiovascular risk factors, and prevalence of coronary artery disease (S2 Table. Patients demographic characteristics).

Cardiac arrest events, in both groups, occurred mostly near the residential environment or at home. In more than half of the patients, OHCA was the first cardiac event documented, without prior ACS events having taken place. About 40% of patients complained of chest pain prior to the event. The median time for ROSC was 30 minutes. Ventricular fibrillation was the most common first-documented rhythm seen in patients treated with hypothermia, whereas asystole and PEA were seen mostly in the non-treated group (68.4% and 58.8%, p < .001, respectively) (Table 1. Preliminary admission data).

In almost 70% of the patients, ischemic changes were observed in the initial ECG. ST elevation was seen in half of the cases. New LBBB was rarely seen in the TH group. Primary percutaneous coronary intervention (PPCI) was done in more than 70% of patients in both groups, and in more than 90% of patients, a coronary abnormality was seen, and infarct-related artery (IRA) was identified. Three vessels disease (TVD) was seen in 40–50% of cases and only 70–80% of PCIs were considered successful (Table 2. ECG, echocardiographic and coronary angiography data).

The occurrence of major cardiovascular events (MACE), including myocardial infarction, heart failure, arrhythmias, cardiac arrest, and sepsis, up to one year after discharge did not differ between the study groups (Table 3. Primary and Secondary end points).

There was no significant difference in mortality rates during hospitalization and up to one year after discharge between the two study groups. Patients who survived OHCA had a high

**Table 1. Preliminary admission data (PAD).**

| | Therapeutic Hypothermia (n = 57) | Non-Therapeutic Hypothermia (n = 35) | p-Value |
|---|---|---|---|
| **Place of the event** | | | **.98** |
| Home | 32 (56.1) | 19 (54.3) | |
| Public place | 19 (33.3) | 12 (34.3) | |
| Medical Clinic [X] | 6 (10.5) | 4 (11.4) | |
| Preceding chest pain [Y] | 22 (38.6) | 16 (45.7) | **.50** |
| Time to CPR (min) [Z] median ± range) | 4.94±8.19 (0; 0–40) | 9.32±11.31 (2.5; 0–35) | **.12** |
| Time for ROSC[S] (min)[W] (median ± range) | 30.48±18.77 (28; 5–75) | 26.77±14.01 (28; 7–50) | **.76** |
| First documented arrhythmia | | | **.01** |
| Ventricular Fibrillation | 39 (68.4) | 14 (41.2) | |
| Asystole/PEA | 17 (29.8) | 20 (58.8) | |

[X] Medical clinic = local clinic with the capability of first care and basic life support assistant

[Y] Missing/Unclear data appears in three patients

[Z] Time to CPR = Duration from cardiac arrest to initiation of cardiopulmonary resuscitation (CPR)

[W] Missing/Unclear data appears in seven patients

[S] ROSC time = Duration from CPR initiation (by bystander or Emergency medical service) to return of spontaneous circulation (ROSC)

mortality rate of about 50% during hospitalization. During the first year after discharge, there was an additional mortality risk of approximately 12% (Table 4. 1-year mortality subgroup analysis).

Sepsis was observed in 51% of patients treated with hypothermia compared to 40% of untreated patients (p < .04). The most common cause for sepsis among both groups was pneumonia. In 65.5% of patients in the TH group, pathogens were identified compared to 35.7% of patients in the Non-TH group. The most common Gram-positive bacteria were *Streptococcus pneumonia*, followed by *Staphylococcus aureus*. *Escherichia coli* and *Hemophilus influenza* were the predominant Gram-negative pathogens (S3 Table. Cause of sepsis among study groups).

Cox regression analysis indicated that the age, primary arrhythmia, and Time to CPR were the highest-impact variables for 1-year mortality. When adjusting Time to CPR to age, it appears that age and initial arrhythmia were the only significant variables for mortality and had no influence on neurological outcomes.

Patients in the TH group had a larger average CPC scale upon discharge compared to the Non-TH group (3.58±1.65 *vs*. 2.63±1.71, p < .05). There was no significant improvement in average neurological outcomes after 1 year in both groups (S1 Fig. Neurological outcome—30 days & 1-year average CPC score among study groups).

## 4. Discussion

Therapeutic hypothermia requires a considerable amount of resources and highly qualified medical staff. It limits medical care during the first 24–48 hours and is accompanied by a significantly high risk of sepsis.

Initial observational studies, in the early 1990s, showed a neurological benefit in reducing intracranial pressure in patients following head trauma. However, years later, some of these researchers contradicted their own findings by failing to prove a direct connection [25–32].

**Table 2. ECG, echocardiographic and coronary angiography data.**

| | Therapeutic Hypothermia (n = 57) | Non-Therapeutic Hypothermia (n = 35) | p-Value |
|---|---|---|---|
| **ECG** | 38 (68.7) | 26 (74.3) | .44 |
| ST segment elevation | 28 (49.1) | 11 (31.4) | .10 |
| ST segment depression | 8 (14.0) | 3 (8.6) | .52 |
| New Left bundle branch block | 3 (5.3) | 8 (22.9) | .02 |
| New Q pathological Wave | 1 (1.8) | 4 (11.4) | .07 |
| Primary PCI [A] | 41 (71.9) | 25 (71.5) | .96 |
| Identified IRA[B] | 34 (97.1) | 22 (91.7) | ,56 |
| TVD [C] | 23 (56.1) | 9 (39.1) | .19 |
| Opened IRA[B] | 35/41 (85.4) | 16/23 (69.6) | .20 |
| LVEF [D] (%) (median ± range) | 29.16 ±23.6 (35; 0–60) | 20.6 ±22.2 (20; 0–60) | .09 |
| LVEF ≤ 40% | 28 (52.8) | 23 (65.7) | .23 |
| New Mitral regurgitation | 15 (28.3) | 10 (28.6) | .98 |

[A] PCI = Percutaneous coronary intervention

[B] IRA = Infarct related artery

[C] TVD = Three vessels disease

[D] LVEF = Left ventricular ejection fraction

In the beginning, the desired depth of cooling or cooling length was not clear. Initial experiments were performed when cooling was set to a core temperature of 28–32˚C. Initial results failed to demonstrate a significant survival benefit, and many studies abandoned this approach. Fifteen years later, this technique was resumed focusing on a modest cooling of 32–34˚C. The results of the first animal trials were promising, with significant survival rates and better neurologic outcomes. Therapeutic hypothermia was then classified into three categories depending on the depth or cooling intensity: 35–32˚C, 31.9–30˚C, and 29.9–28˚C, or Mild, Modest, and Deep hypothermia, respectively [32].

In 2002, two randomized trials, from Melbourne, Australia and Europe (multicenter), showed a better neurological outcome and higher survival rate in patients treated with mild therapeutic hypothermia. The larger of the two trials involved 136 patients in several European countries and was administered by a research group comparing unconscious patients with a self-pulse capability who survived cardiac arrest outside the hospital when the first arrhythmia observed was ventricular fibrillation. Survival after 6 months of hospitalization was higher in patients treated with mild hypothermia compared with those who were not treated (41% and 55%, respectively, p = .02). The neurological damage upon discharge was more severe among patients who were not treated with hypothermia (39% and 55% for good neurologic outcomes, respectively, p = .002). Similar results were described in the Australian study [33–35].

Our study retrospectively analyses the treatment efficacy during two different time periods, prior to the TH era and afterward, with a time interval of almost two decades between the first and last patient. Despite the tremendous change in primary PCI catheterization, treatment with therapeutic hypothermia has not demonstrate a significant benefit in terms of both survival rate and neurological outcomes in patients.

Post-hoc analysis indicates that selective patients had higher survival rate benefited more following therapeutic hypothermia. Patients admitted due to ventricular fibrillation had a higher survival rate following therapeutic treatment regardless of their age, while patients admitted due to non-shockable rhythm, such as asystole, experienced a modest benefit only if

**Table 3. Primary and Secondary end points.**

| | Therapeutic Hypothermia (n = 57) | Non-Therapeutic Hypothermia (n = 35) | p-Value |
|---|---|---|---|
| **CPC[1] on discharge** | | | .39 |
| 1 | 13 (22.8) | 7 (43.8) | |
| 2 | 3 (5.3) | 1 (6.3) | |
| 3 | 6 (10.5) | 3 (18.8) | |
| 4 | 8 (14.0) | 1 (6.3) | |
| 5 | 27 (47.4) | 4 (25.0) | |
| CPC (average ± SD)–at discharge | 3.58±1.65 | 2.63±1.71 | .05 |
| CPC (average ± SD)–at 30 days | 3.26±1.78 | 2.38±1.59 | .07 |
| CPC (average ± SD)–at 1 year | 3.28±1.84 | 2.08±1.55 | .03 |
| In-hospital mortality | 26 (45.6) | 19 (54.3) | .42 |
| 30 days mortality | 0 (0.0) | 0 (0.0) | — |
| 1-year mortality | 2 (6.4) | 3 (18.8) | .32 |
| **MACE [2] –at 30 days** | 1/31 (3.2) | 0 (0.0) | >.99 |
| Myocardial Infarction | 0 (0.0) | 0 (0.0) | — |
| Congestive heart failure | 0 (0.0) | 0 (0.0) | — |
| Ventricular arrhythmia | 1 (3.2) | 0 (0.0) | >.99 |
| Cardiac arrest | 0 (0.0) | 0 (0.0) | — |
| Sepsis | 1 (3.2) | 0 (0.0) | >.99 |
| **MACE[2] –at 1-year** | 7 /31 (22.6) | 1/16 (6.2) | .23 |
| Myocardial Infarction | 1 (3.2) | 0 (0.0) | >.99 |
| Congestive heart failure | 0 (0.0) | 0 (0.0) | — |
| Ventricular arrhythmia | 3 (9.7) | 1 (6.2) | >.99 |
| Cardiac arrest | 1 (3.2) | 0 (0.0) | >.99 |
| Sepsis | 4 (12.9) | 0 (0.0) | .28 |

[1] CPC = Cerebral performance category scale

[2] MACE = Major adverse cardiovascular events

they were younger than 65 years. Hypothermic treatment in patients older than 65 years who are admitted due to asystole may be at risk of harm from this treatment (Fig 2. Preliminary admission data).

Our study did not reveal a significant benefit in neurological outcomes from therapeutic treatment. In our estimation, this could be explained by various factors such as small sample size and lack of an accurate and more specified assessment model to asses neurological status.

**Table 4. 1-year mortality subgroup analysis (arranged by survival rate).**

| Group | N | Mean Days Survived | Standard Error of the mean | 95% CI (days) |
|---|---|---|---|---|
| Age≥65 and VF[1] | 8 | 228.5 | 47.6 | 135.2–321.8 |
| Age<65 and VF[1] | 12 | 208.4 | 33.7 | 142.3–274.6 |
| Age<65 and asystole | 27 | 191.3 | 55.0 | 83. 6–299.0 |
| Control group | 10 | 153.0 | 28.8 | 96.5–209.5 |
| Age≥65 and asystole | 35 | 72.0 | 46.0 | 0.0–162.3 |
| Overall | 92 | 177.4 | 18.2 | 141.8–213.0 |

[1] VF = Ventricular Fibrillation

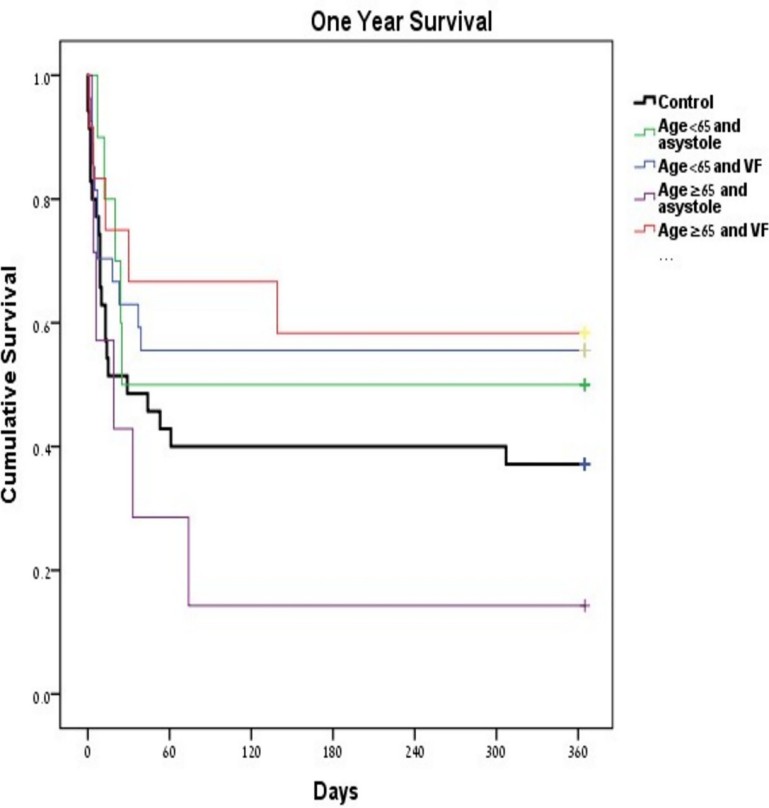

**Fig 2. 1-year survival rate among subgroup (K-M graph).**

We recommend designing a model of targeted patient selection for TH based on data from multiple experienced centers and validating it prospectively. Additionally, we recommend adopting or designing a more accurate and detailed neurological assessment model for cardiac arrest patients.

### 4.1 Limitations of the study

This is a retrospective study using data spanning a long period of time. Data was taken from computer systems without having the ability to assess their reliability. There is a natural bias in the choice of patients admitted to the cardiac unit in our center, with a high likelihood of suffering from cardiac etiology.

In addition, we used two different time periods that are expressed by different baseline characteristics. This bias was reduced by propensity score matching, yet differing preliminary admission data may still affect the research outcome, e.g., prolonged ischemia time, incidence rates of left bundle branch block, revascularization success rates, etc.

## 5. Conclusions of the study

Therapeutic hypothermia carries a considerable risk of mortality and morbidity. The use of Therapeutic Hypothermia may not be suitable for all patients and in all cases of out-of-hospital cardiac arrest. Our observation and post-hoc analysis suggest that proper patient selection, based on initial presented arrhythmia and age of the patient, had a significant impact on 1 year mortality. Our research may explain the different results of previous studies.

## Supporting information

**S1 Table. Inclusion and exclusion criteria.**
(DOCX)

**S2 Table. Patients demographic characteristics.**
(DOCX)

**S3 Table. Cause of sepsis among study groups.**
(DOCX)

**S1 Fig. Neurological outcome– 30 days and 1-year average CPC score among study groups.**
(DOCX)

**S1 Dataset.**
(XLSX)

## Acknowledgments

We would like to thank Ediqo.com for their manuscript editing & proofreading service.
We would like to thank Dr. Regina Koren for her assistance with data analysis.

## Author Contributions

**Conceptualization:** Ofir Koren, Yoav Turgeman.

**Data curation:** Ofir Koren, Sawsan Yosefia.

**Formal analysis:** Ofir Koren, Sawsan Yosefia.

**Funding acquisition:** Ofir Koren, Sawsan Yosefia.

**Investigation:** Ofir Koren, Sawsan Yosefia.

**Methodology:** Ofir Koren, Sawsan Yosefia.

**Project administration:** Ofir Koren, Sawsan Yosefia.

**Resources:** Ofir Koren, Yoav Turgeman.

**Software:** Ofir Koren, Yoav Turgeman.

**Supervision:** Ofir Koren, Ehud Rozner, Yoav Turgeman.

**Validation:** Ofir Koren, Ehud Rozner, Yoav Turgeman.

**Visualization:** Ofir Koren, Ehud Rozner, Yoav Turgeman.

**Writing – original draft:** Ofir Koren, Ehud Rozner, Sawsan Yosefia, Yoav Turgeman.

**Writing – review & editing:** Ofir Koren, Ehud Rozner, Sawsan Yosefia, Yoav Turgeman.

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
