## [Decision Letter · Decision Letter 0]

9 Oct 2019

PONE-D-19-24905

Proper patient selection for Therapeutic Hypothermia after Out of Hospital Cardiac Arrest improve 1-year Survival rate

PLOS ONE

Dear Dr. Koren,

Thank you for submitting your manuscript to PLOS ONE. After careful consideration, we feel that it has merit but does not fully meet PLOS ONE’s publication criteria as it currently stands. Therefore, we invite you to submit a revised version of the manuscript that addresses the points raised during the review process.

We would appreciate receiving your revised manuscript by Nov 23 2019 11:59PM. To enhance the reproducibility of your results, we recommend that if applicable you deposit your laboratory protocols in protocols.io, where a protocol can be assigned its own identifier (DOI) such that it can be cited independently in the future. For instructions see: http://journals.plos.org/plosone/s/submission-guidelines#loc-laboratory-protocols

We look forward to receiving your revised manuscript.

Kind regards,

Theodoros Xanthos

Academic Editor

PLOS ONE

Journal Requirements:

1. We note that you have stated that you will provide repository information for your data at acceptance. Should your manuscript be accepted for publication, we will hold it until you provide the relevant accession numbers or DOIs necessary to access your data. If you wish to make changes to your Data Availability statement, please describe these changes in your cover letter and we will update your Data Availability statement to reflect the information you provide.

2. Please remove your figures from within your manuscript file, leaving only the individual TIFF/EPS image files, uploaded separately.  These will be automatically included in the reviewers’ PDF.

Reviewers' comments:

Reviewer's Responses to Questions

**Comments to the Author**

1. Is the manuscript technically sound, and do the data support the conclusions?

Reviewer #1: Partly

Reviewer #2: Yes

2. Has the statistical analysis been performed appropriately and rigorously? 

Reviewer #1: Yes

Reviewer #2: Yes

3. Have the authors made all data underlying the findings in their manuscript fully available?

Reviewer #1: Yes

Reviewer #2: Yes

4. Is the manuscript presented in an intelligible fashion and written in standard English?

Reviewer #1: No

Reviewer #2: No

5. Review Comments to the Author

Reviewer #1: Thank you very much for giving me the opportunity to review this manuscript. In the present study Koren et al. retrospectively examined whether proper selection of cardiac arrest patients would increase the effectiveness of therapeutic hypothermia (TH) in terms of survival and neurological outcome. They concluded that patients treated with TH following VF benefited the most (in term of 1-year survival), compared to the modest benefit showed in patients under 65 with asystole, which is an interesting finding. No effect was demonstrated regarding neurological outcome. However, the following issues have arisen:

1. The authors report that they included only OHCA patients in the study. Nonetheless, in table 1 it is clear that some patients suffered from cardiac arrest inside a medical facility. Please clarify this in the manuscript.

2. It is not clear at what temperature TH was performed. Please clarify this in the manuscript.

3. Please use the abbreviations that you have already defined, all over the manuscript and define the ones that have not been explained. Also, define all abbreviations used in tables and figures.

4. Sustained VT is not a cardiac arrest rhythm. Pulseless VT is a shockable rhythm. Please change this in the “introduction” section.

5. Due to the grammar and language mistakes in the entire article, I strongly recommend manuscript editing by a native speaker.

6. A huge part of the “introduction section” is a review of the current literature. Please move this part to the “discussion section”.

7. Clarify what “ischemic time” means and how it was measured.

8. All patients were unresponsive after ROSC or did they have a good level of consciousness when they arrived at your hospital?

9. The patients were treated for OHCA with AEDs? How did the authors know the presenting rhythm (PEA, asystole, VF, PVT)? Please include this information in the manuscript.

10. Change sudden cardiac arrest to “cardiac arrest”.

11. Many statements are not followed by a reference. Please insert all necessary references.

12. In the Introduction section you refer to 3 categories of TH depending on the depth of cooling intensity but then you state only 2 categories. Please report all 3.

13. From all patients included in TH and non-TH groups you had information about the preceding symptoms, ischemic time, time for ROSC etc? If not, include the exact number of patients from whom you had information in table 1. Also state what the number in parenthesis represents.

14. Please define what “median time for ROSC” represents. Is it the time from the beginning of CPR by bystanders, the EMS or from the cardiac arrest?

Reviewer #2: Dear authors,

I read with interest your study which examined whether proper selection of patients increase the efficacy of TH and reported that it failed to demonstrate a significant benefit in neurological outcome. Of course, this is a retrospective study with the well-known limitations associated with this type of studies.

Due to ethical issues, I would be very reluctant to withhold TH in any patient admitted in my Department and I would need very strong evidence before I do it. Your study can help the resuscitation community by being used as the first step for conducting large RCTs on proper patient selection. I have some major comments:

1. The issue of proper patient selection is paramount. Of note, there are published studies assessing the effect of BMI in patients who are not treated with TH and adding this information (BMI of your patients) would be important.

2.. The non-significant benefit of TH is a result that cannot be neglected and merits further research. Notably, there was a small difference between the two groups, which could be significant if the study sample was larger. Also, your study confirms the adverse events associated with TH that have been reported by other authors and RCTs, e.g. infection, but you do not describe the method for inducing TH in your institution (initiated in ED/ICU, blankets/IV fluids, etc). Please comment.

3. Some minor comments: Your introduction section must be organized to 3-4 smaller paragraphs. It is too large. Also, please add the matching process (in-detail) to the supplementary material and have the manuscript reviewed by a native English speaker.

6. PLOS authors have the option to publish the peer review history of their article (what does this mean?). If published, this will include your full peer review and any attached files.

Reviewer #1: No

Reviewer #2: No

---

## [Author Response · Author response to Decision Letter 0]

17 Oct 2019

A separate file labeled ""response to reviewers" was uploaded.

---

## [Editor Report · Decision Letter 1]

25 Nov 2019

PONE-D-19-24905R1

Proper patient selection for Therapeutic Hypothermia after Out of Hospital Cardiac Arrest improve 1-year Survival rate

PLOS ONE

Dear Dr. Koren,

Thank you for submitting your manuscript to PLOS ONE. After careful consideration, we feel that it has merit but does not fully meet PLOS ONE’s publication criteria as it currently stands. Therefore, we invite you to submit a revised version of the manuscript that addresses the points raised during the review process.

Specifically, please address the following editorial requests:

- In your manuscript you state that "Informed consent was not required due to the confidentiality of patent data" and also "researchers contacted the patient or close relatives for further details". If the data was kept confidential please explain how patients or their relatives were contacted.

- In your Methods section, please provide additional details regarding your statistical analyses. A sample size calculation is referred to but not described, nor an appropriate sample size stated. Also, no post-hoc corrections for multiple comparisons are mentioned in your manuscript. If these were performed please include them in the text, or justify their absence.

- Please revise parts of your manuscript (and title) that use the phrase "proper patient selection". This is unclear and open to misinterpretation in the context of whether adequate clinical practice was followed for the patients included in the study, can you therefore revise the title and references to this in the text to ensure it aligns to the analysis undertaken as part of the study. Additionally, avoid referring to any "causal" links in the text. The retrospective nature of this study does not allow for these to be made, please revise references to ‘effectiveness’ or ‘efficacy’ and tone down your 'Conclusions' section accordingly.

We would appreciate receiving your revised manuscript by Jan 06 2020 11:59PM. Please include the following items when submitting your revised manuscript:

We look forward to receiving your revised manuscript.

Kind regards,

Natasha Rickett

Associate Editor

PLOS ONE

On behalf of,

Theodoros Xanthos

Academic Editor

PLOS ONE

---

## [Author Response · Author response to Decision Letter 1]

27 Nov 2019

response to reviewers file was uploaded

---

## [Editor Report · Decision Letter 2]

11 Dec 2019

Therapeutic Hypothermia after Out of Hospital Cardiac Arrest improve 1-year Survival rate for selected patients

PONE-D-19-24905R2

Dear Dr. Koren,

We are pleased to inform you that your manuscript has been judged scientifically suitable for publication and will be formally accepted for publication once it complies with all outstanding technical requirements.

With kind regards,

Theodoros Xanthos

Academic Editor

PLOS ONE
---

## [Editor Report · Acceptance letter]

23 Dec 2019

PONE-D-19-24905R2 

Therapeutic Hypothermia after Out of Hospital Cardiac Arrest improve 1-year Survival rate for selected patients 

Dear Dr. Koren:

I am pleased to inform you that your manuscript has been deemed suitable for publication in PLOS ONE. Congratulations! Your manuscript is now with our production department. 

With kind regards,

on behalf of

Professor Theodoros Xanthos 

Academic Editor

PLOS ONE